# A Video-Based Technique for Heart Rate and Eye Blinks Rate Estimation: A Potential Solution for Telemonitoring and Remote Healthcare

**DOI:** 10.3390/s21051607

**Published:** 2021-02-25

**Authors:** Vincenzo Ronca, Andrea Giorgi, Dario Rossi, Antonello Di Florio, Gianluca Di Flumeri, Pietro Aricò, Nicolina Sciaraffa, Alessia Vozzi, Luca Tamborra, Ilaria Simonetti, Gianluca Borghini

**Affiliations:** 1Department of Anatomical, Histological, Forensic and Orthopaedic Sciences, Sapienza University, 00185 Rome, Italy; alessia.vozzi@uniroma1.it (A.V.); tamborra.1230281@studenti.uniroma1.it (L.T.); simonetti.1522496@studenti.uniroma1.it (I.S.); 2BrainSigns srl, 00185 Rome, Italy; andrea.giorgi@brainsigns.com (A.G.); antonello.diflorio@brainsigns.com (A.D.F.); gianluca.diflumeri@brainsigns.com (G.D.F.); pietro.arico@brainsigns.com (P.A.); nicolina.sciaraffa@brainsigns.com (N.S.); 3Department of Business and Management, LUISS University, 00197 Rome, Italy; dario.rossi@brainsigns.com; 4Department of Molecular Medicine, Sapienza University of Rome, 00185 Rome, Italy; 5IRCCS Fondazione Santa Lucia, 00179 Rome, Italy; 6People Advisory Services Department, Ernst & Young, 00187 Rome, Italy

**Keywords:** facial video, healthcare, telemedicine, neurophysiological assessment, signal processing, heart rate, eye blinks, mental states evaluation

## Abstract

Current telemedicine and remote healthcare applications foresee different interactions between the doctor and the patient relying on the use of commercial and medical wearable sensors and internet-based video conferencing platforms. Nevertheless, the existing applications necessarily require a contact between the patient and sensors for an objective evaluation of the patient’s state. The proposed study explored an innovative video-based solution for monitoring neurophysiological parameters of potential patients and assessing their mental state. In particular, we investigated the possibility to estimate the heart rate (HR) and eye blinks rate (EBR) of participants while performing laboratory tasks by mean of facial—video analysis. The objectives of the study were focused on: (i) assessing the effectiveness of the proposed technique in estimating the HR and EBR by comparing them with laboratory sensor-based measures and (ii) assessing the capability of the video—based technique in discriminating between the participant’s resting state (*Nominal condition*) and their active state (*Non-nominal condition*). The results demonstrated that the HR and EBR estimated through the facial—video technique or the laboratory equipment did not statistically differ (*p* > 0.1), and that these neurophysiological parameters allowed to discriminate between the Nominal and Non-nominal states (*p* < 0.02).

## 1. Introduction

Nowadays telemedicine platforms are employed in a wide range of medical and clinical applications, such as the diabetes management [1], asthma monitoring [2,3], chronic disease [4,5] and age-related diseases [6,7]. According to Armaignac and colleagues [8], telemedicine is also applied in critical care [9,10] to overcome the increasing patient demands and shortage of intensivists, issues that may occur in different contexts, first and foremost during the COVID-19 pandemic [11]. Telemedicine could be defined as the use of technological equipment to provide a clinical and medical assistance when a physical distance separates patients and providers. Telemedicine also includes managing patients through monitoring devices controlled by physicians and nurses in remote locations [12], internet-based video-conferencing platforms for communicating with patients remotely [13], and asynchronous and synchronous systems for providing clinical care through the use of wearable devices [14,15]. Besides the therapeutical applications, telemedicine is also employed for remote monitoring of patients. The objective of this passive branch of telemedicine is to provide a warning to clinicians or doctors when the neurophysiological and physiological data collected from the patients indicate an adverse clinical event [16,17]. Several studies have already demonstrated the effectiveness of telemedicine in improving patients’ outcomes [18,19], while other studies have showed its benefits in terms of hospitalization reduction, a crucial aspect especially during a severe pandemic such as the COVID-19 one [11].

However, all the above mentioned telemedicine and remote healthcare concepts require physical contact between the patient and sensors, and the need of high—qualified personnel to set up the entire equipment and provide technical assistance to the patients. The internet-based video-conferencing telemedicine platforms do not require physical contact between patients and providers, although they imply a large limitation due to the lack of sensors to evaluate the neurophysiological parameters of the patients.

The present study explored an innovative approach for the telemedicine and telemonitoring that aims at estimating the heart rate (HR) and the eye blinks rate (EBR) through the analysis of the patient’s face video recorded by mean of a webcam. This video—based technique does not require any technical support to perform the measurements, as it does not require a physical contact between the user and the sensors. Furthermore, this kind of methodology is not expensive as the actual technologies, i.e., medical and wearable devices, employed in telemedicine and remote healthcare. Besides the clinical implications related to the HR monitoring, previous work demonstrated how this neurophysiological parameter is involved in human mental states assessment like the mental workload [20,21,22]. Similarly, the EBR is associated with specific mental states like the visual attention [23]. In fact, it was demonstrated that a decrease of EBR corresponds to greater processing of information [24]. Such two aspects indicate the suitability of HR and EBR parameters for characterizing the patient’s mental states in terms of attention and mental workload. Video-based techniques imply the recording of the patient’s facial video consequently they cannot be applied on patients who are not in front a video camera. The proposed technique for HR evaluation was already explored in prior works with promising results [22,25,26], and it is based on the modulation of the reflected ambient light from the skin by the absorption spectrum of hemoglobin in the patient’s blood [25]. In other words, such analysis is based on the extraction and processing of the Red component of the patient’s facial video. The minute—color variations on the skin are created by blood circulation, and they module the Red component of the video signal along the time. The remote EBR monitoring by means of facial video analysis was explored in recent works too. Zhang and colleagues [26] demonstrated the reliability of multi-channel ICA to detect eye blinks from smartphone facial videos, while Tsujikawa in 2018 [27] evaluated the reliability of EBR estimation from 30 frame per second (fps) facial video cameras. In this regard, the first objective of the present study was to investigate the reliability of the video—based technique for the simultaneous HR and EBR estimation. These neurophysiological parameters were compared with the corresponding ones computed from the electrocardiographic (ECG) and electrooculographic (EOG) signals gathered through laboratory equipment. Secondly, the experimental protocol was designed to represent the situation in which the patient’s state deviates from a resting condition. The deviation from the resting condition (*nominal condition*) could play a crucial role in several telemedicine application, such as the sleep apnea remote monitoring [28,29,30] and cardiovascular diseases remote monitoring while sleeping [31], but also in operative applications involving narcoleptic patients [32] and in emotional states discrimination, since several previous works demonstrated how the HR and EBR parameters are involved in the emotional state modulation [33,34]. The video–based technique has great potential in this latter application, especially in isolation and health emergency situations, a relevant risk factor for pathologies such as depression, anxiety and stress [35,36]. Therefore, the present study explored and validated the video-based method in terms of neurophysiological parameters estimation, i.e., the HR and EBR, with respect to conventional sensors. In summary, the present work aimed at addressing the following two experimental questions:Is the considered video-based technique reliable in terms of HR and EBR estimation?Is the considered video-based technique capable of discriminating between a *nominal* and a *non-nominal* state of the patient?

## 2. Materials and Methods

### 2.1. Participants

Informed consent for study participation, publication of images, and to use the video material were obtained from a group of 15 students, eight males and seven females (30.6 ± 3.7 years old) from the Sapienza University of Rome (Italy) after the explanation of the study. The experiments were conducted following the principles outlined in the Declaration of Helsinki of 1975, as revised in 2000. The study protocol received the favorable opinion from and has been approved by the Ethical Committee of the Sapienza University of Rome (protocol n. 2507/2020 approved on the 04/08/2020). The study involved only healthy participants, recruited on a voluntary basis. Furthermore, the students were free to accept or not to take part to the experimental protocol, and all of them have accepted to participate to the study. Only aggregate information were released while no individual information were or will be diffused in any form.

### 2.2. Experimental Protocol

To simulate the switch between a *n**ominal* and a *non-nominal* state in this experimental protocol, three tasks were designed:The n-Back (NB) task. A well-known computer-based psychological test used to manipulate workload, or more specifically working memory load [37]. Within this task a sequence of stimuli is presented to the user. The goal is to indicate when the current stimulus matches the stimulus that occurred in the series *n* steps before. The factor *n* can be adjusted to make the task more difficult or easier. A baseline and three conditions (0-back, 2-back, and 2-back stressful) of such task were tested in the proposed study, all of them with different levels of difficulty. In all conditions, 21 uppercase letters were used, which were displayed for 500 ms and an inter-stimulus interval randomized between 500 to 3000 ms; 33% of the displayed letters were target. During the baseline (1 min duration), the same 21 uppercase letters was presented to the participants with no interaction required.The Doctor Game (DG). The aim of the game was to remove small objects from the board without touching the edges. Here, a baseline and three difficulty levels were tested too.Two interactive web calls (WEB) were performed. Three conditions of such task were performed: (i) Baseline condition, in which the participants looked at the web platform interface without reacting; (ii) Positive condition, in which the test persons were asked to report the happiest memory of their life; (iii) Negative condition, in which the test persons were asked to report the saddest memory of their life.

The participants went under training phases before performing each different task in order to avoid habituation bias. Considering the two main objectives of the present work, the different difficulty levels of the experimental tasks were not considered in the analysis. In particular, the neurophysiological parameters evaluated during the resting state, in which the participant rested in front of the PC screen, were referred to the *Nominal* condition while the neurophysiological parameters evaluated during the remaining experimental conditions, averaged along such conditions within each task, were referred to the *Non-nominal* condition.

### 2.3. Questionnaires

To validate the neurophysiological results two kinds of questionnaires were used, which were filled in after each experimental condition. The questionnaires were explained at the beginning of the experiment and the participants were trained to fill them before starting with the experiments. The following questionnaires were selected:Self-assessment Manikin (SAM), consisting in a picture-oriented questionnaire [38] developed to measure the valence/pleasure of the response (from positive to negative), perceived arousal (from high to low levels), and perceptions of dominance/control (from low to high levels) associated with a person’s affective reaction to a wide variety of stimuli. After each experimental condition the participants were asked to provide only three simple judgments along each affective dimension (on a scale from 1 to 9) that best described how they felt during the condition just executed. This questionnaire was selected to have a subjective indication about the current state of the participants in terms of pleasure, arousal and control with the respect of each experimental condition of WEB task.NASA Task Load Index (NASA-TLX), consisting of six sub-scales representing independent groups of variables: mental, physical and temporal demands, frustration, effort and performance. The participants were initially asked to rate on a scale from “low” to “high” (from 0 to 100) each of the six dimensions during the task. Afterwards, they had to choose the most important factor along pairwise comparisons [39]. The NASA-TLX was selected for subjectively quantify the mental demand perceived by the participants with the respect of the experimental condition of the DG and NB tasks.

### 2.4. Eye Blinks Signal Recording and Analysis

The EBR information were obtained by estimating the vertical electrooculographic (EOG) activity from a traditional electroencephalography (EEG) channel [R] [40]: the activity was recorded between a gel-based Ag/AgCl electrode placed on the participant’s Fpz scalp location (Figure 1) and reference electrodes placed on the earlobe, connected to the BEMicro system (EBNeuro, Firenze, Italy) with a sampling frequency of 256 (Hz). Firstly, the signal was band-pass filtered using a 5th order Butterworth filter within the frequency range of 2–10 Hz. In this way the recorded signal can be considered as an estimation of the vertical EOG one. The eye blinks detection method was performed in two main steps:(i)Threshold calculation(ii)Pattern Matching.

In (i) the Eyes Open condition was used to identify a threshold that when exceeded identified a potential blink. The threshold was calculated as follows, according with the BLINKER algorithm [41]:(1)Threshold=mean(EOG Eyes Open)+3∗robustStdDev
where robustStdDev is the mean absolute deviation of the corresponding EOG channel. In (ii), every time the EOG signal exceeded the computed threshold, the Pearson correlation between a common blink template and the EOG signal was computed within each experimental condition. If this value was higher than 0.9, a potential blink would be classified as “real blink”. The EBR feature estimated for each participant in each condition was calculated as the mean of the total number of blinks in every condition per minute.

Regarding the EBR estimation through the video-based technique, a PC webcam (Microsoft, Albuquerque, New Mexico, USA) was used for facial video recording during the experimental protocol (Figure 1).

The RGB camera was set to a resolution of 640 × 480 (pixel) at a frame rate of 30 (fps). The camera was placed in front of the participant. Subsequently, the recorded video was analyzed offline. The participant’s face was automatically identified using a specific Python library named *Dlib* [42] coupled with an *adaBoost* classifier [43]. In particular, such library allowed us to select 68 facial features. Subsequently, the positions of the participant’s eyelids were identified frame by frame. The distance between the inferior and the superior eyelids was computed for both the eyes [44]. Then, such discrete signal was filtered between 1 and 3 (Hz) for noise removal, and a threshold was computed as the quadratic mean of the signal along each specific experimental condition [45]. Each event exceeding such a threshold was finally classified as eye blink. Here too, the EBR parameter was computed as the mean of the total number of blinks in every condition per minute. The required processing time for computing one EBR value was 0.174 s. The main steps of the described video—signal processing for EBR estimation are presented in Figure 2.

### 2.5. ECG Signal Recording and Analysis

The ECG signal was gathered by means of gel-based Ag/AgCl electrode fixed on the participant’s chest (Figure 1), connected to the BEMicro system and referred to the potential recorded at both the earlobes, with a sampling frequency of 256 Hz. First, the ECG signal was filtered using a 5th-order Butterworth band-pass filter (1–4 Hz) in order to reject the continuous component and the high-frequency interferences, such as that related to the mains power source. At the same time, the purpose of this filtering was to emphasize the QRS process of the ECG signal [46,47,48]. The following step consisted in computing the ECG signal to the power of 3 to emphasize the heartbeat peaks, as they generally have the higher amplitude, and at the same time reduce spurious artefacts peaks. Finally, we measured the distance between consecutive peaks (i.e., each R peak corresponds to a heartbeat) in order to estimate the heart rate (HR) values every 60 s. 

Regarding the HR estimation by means of the video–based technique, the same participants’ facial video was analyzed. As described for the EBR estimation, the 68 visual feature required for the facial recognition were selected using the *Dlib* Python library [42] in conjunction with the *adaBoost* classifier [43]. This classifier was employed for the automatic face detection and it was based on the YCbCr Color model [49,50], in order to perform the face detection according with the luminance and chrominance variations of the video. First, the Red (R) component was selected and extracted from the raw signal, through the application of the fast Fourier transform (FFT) and principal component analysis (PCA). The PCA algorithm was also applied for fluctuations removal from the R component, technically implemented in the *sklearn.decomposition.PCA* Python library included in the Scikit-Learn Python library [51]. The considered signal was gathered from the participant’s cheeks, in each image frame, referenced to the participant’s eyes and nose [52]. Then, the clean R component was detrended for illumination variations compensation, by mean of the method proposed by Tarvainen and colleagues [53] based on smoothness priors technique employing a smoothing parameter λ = 10 and a cut-off frequency = 0.060 Hz. Subsequently, Hamming filtering (128 point, 0.6–2.2 Hz) was applied to the R detrended component. Finally, the filtered signal was normalized using z-score [54] by the formula provided below:(2)$$X_{i}=\;\frac{Y_{i}\left(t\right)-\;\mu_{i}\left(t\right)}{\delta_{i}}$$

The HR values were computed with a 60 s time resolution for each experimental condition, considering a sliding time window of 100 image frames for each HR value. The processing time for computing one HR value was 0.041 s. The main steps of the described video—signal processing for HR estimation are presented in Figure 3.

### 2.6. Statistical Analysis

All the considered neurophysiological parameters, i.e., the EBR and HR, were normalized to obtain comparable distributions related to each sensor technology employed in the study. The normalization consisted in the subtraction of the baselines from the respective values estimated during each experimental condition. The statistical analysis was performed on the normalized parameters. The Shapiro–Wilk test was performed to determine the normality of each distribution involved in the analyses. The Student’s t-test was used to compare normal pairs, while the Wilcoxon signed-rank test was performed if the normality was not confirmed. For all tests, the statistical significance was set at α = 0.05.

## 3. Results

The results related to the DG task will not be reported because almost all the participants got too close to the game board to accurately extract the objects causing face-video signal loss therefore the impossibility to acquire and consequently analyze their facial videos.

### 3.1. Methodology Comparison

Regarding the EBR estimation, the paired Wilcoxon signed-rank test performed on the normalized EBR (*EBR’*) evaluated during the NB and WEB tasks did not show any significant difference (Figure 4) between the video—based technique and the laboratory technology (NB: *p* = 0.7; WEB: *p* = 0.5). The percentage difference between the EBR estimated through the video—based technique and the laboratory equipment was 4.5% during the NB task and 4.8% during the WEB task.

The same result (Figure 5) was observed on the paired Wilcoxon signed-rank test performed on the normalized HR (*HR’*) estimated during the NB and WEB tasks (NB: *p* = 0.2; WEB: *p* = 0.4). The percentage difference between the HR estimated through the video–based technique and the laboratory equipment was 9.3% within the NB task and 3.3% within the WEB task.

Furthermore, to investigate the reliability of the video-based technique with respect to the laboratory technology, the repeated measure correlation (rmcorr) analysis was performed. As reported in Figure 6, the rmcorr analysis [55] performed between the EBR estimated by the laboratory and video-based technique every 60 s showed a positive (R = 0.73) and significant (*p* < 10^−27^) correlation, as demonstration of how the two technology provided similar EBR estimations. Similarly, Figure 6 shows a positive (R = 0.64) and significant (*p* < 10^−18^) correlation between the HR values estimated every 60 s by means of the laboratory and the video-based technique.

### 3.2. Mental States Discrimination

The results of the Wilcoxon signed-rank test performed on the NASA-TLX reported significant (*p* = 0.0005) differences among the *nominal* and the *non-nominal* conditions of the NB task in terms of perceived mental demand (Figure 7). Similarly, the Wilcoxon signed-rank test performed on the SAM questionnaire demonstrated a significant (*p* = 0.02) increase of the perceived arousal and control between the Nominal and the Non-nominal conditions of the WEB task (Figure 7).

As mentioned in the Introduction, the second objective of the present study consisted in assessing the capability of the video-based technique in discriminating the participants’ state while they were in a resting state (*nominal*) or in an active state (*non-nominal*). Regarding the NB task, the paired Wilcoxon signed-rank test performed on the normalized EBR and HR estimations provided by the video-based technique (Figure 8) showed a significant difference between the *nominal* and *non-nominal* conditions (EBR: *p* = 0.0002; HR: *p* = 0.03).

Similarly, the paired Wilcoxon signed-rank test performed on the normalized EBR and HR evaluated by the video-based technique during the WEB task (Figure 9) showed a significant difference between the *nominal* and *non-nominal* conditions (EBR: *p* = 0.0003; HR: *p* = 0.02).

## 4. Discussion

The present study aimed at investigating the reliability of an innovative video—based technique in estimating neurophysiological parameters (i.e., EBR and HR) while dealing with different activities to find out if it could be a potential solution for healthcare telemonitoring of patients. Regarding the NB and WEB tasks, the results demonstrated the reliability of the video—based technique compared with the laboratory technology, generally considered as the gold—standard in scientific literature [56]. Moreover, the repeated measure correlation analysis revealed that the video—based technique was able to capture the considered neurophysiological parameters’ dynamics with the same capability exhibited by the laboratory device. More importantly for future applications, the statistical analyses demonstrated the capability of the explored video-based technique in discriminating between the *nominal* and *non-nominal* participant’s mental states. In particular, the normalized EBR (EBR’) estimated within the NB and WEB tasks significantly decreased during the *non-nominal* condition, while the normalized HR (HR’) significantly increased during the *non-nominal* condition within both tasks. These evidences are consistent with prior related works. In fact, Aricò and colleagues demonstrated the link between the EBR decrease and the visual attention increase [57], while we already observed in a previous study the relationship between the HR increase and the mental workload increase [22]. With respect to the two experimental tasks, subjective measures, i.e., the NASA-TLX and SAM, demonstrated that the *nominal* and *non-nominal* conditions were actually different in terms of mental demand, therefore validating the experimental hypothesis at the basis of the presented analysis. Such evidences open the path to apply video–based techniques for healthcare monitoring of patients in remote locations. In fact, such a technique does not require any physical contact between the patient and the sensor, nor the presence of a doctor or a facilitator for the sensors setting. In addition, the video-based technique implies very limited costs, since it needs only a commercial webcam, compared to the existing telemedicine platforms, which include commercial and medical wearable devices. Beyond the telemedicine and remote healthcare applications, the explored video–based technique could provide a valuable contribution in operative and industrial applications. To this regard, different works [58,59] already investigated the possible algorithms to automatically discriminate between the condition in which the operator is active and learning, or the one in which the operator is resting, a crucial aspect to trigger the activation of the artificial intelligence (AI) or support system platforms. Moreover, the presented results demonstrated the sensibility of the video–based technique for EBR and HR estimations to the visual attention and mental demand increases. Therefore, such a technique could offer relevant performance operative applications where it is required the minimum interference between the subjects and the sensors [60], in air traffic controllers’ (ATCOs) mental workload and attention evaluations [61] and in car driver’s monitoring [62].

### Limitations

Despite the promising results we should highlight some limitations. The proposed video–based technique implies a direct visual contact between the subject and the video recorder, a condition that could not be easy to achieve in specific context as the telemedicine one, where the patient could not stand in front of a camera for long time period. In fact, during the execution of the DG task the posture of almost all the participants did not allow to acquire the participant’s face, hence to neither estimate the neurophysiological parameters considered nor assess the participants’ mental states. Therefore this aspect should be carefully considered when a video–based solution would be employed. Moreover, the investigated video–based technique could likely be sensible to illumination variations [63], a parameter that is not always controllable. Such a limitation could be solved or at least mitigated by using a camera featuring automatic brightness regulation for the facial video recording. Finally, it has to be noted that the video—based technique requires specific sensing and processing times, depending on the chosen sliding time window to perform the measurements among the image frames and on the PC used for the analysis.

## 5. Conclusions

The proposed study demonstrated the reliability of the innovative video–based technique for computing the EBR and HR neurophysiological parameters. Both parameters evaluated through the video–based technique did not differ by more than 5%, except for the HR evaluated during the NB task which differed by 9.3%, from the measurements provided by laboratory equipment. In addition, the results revealed its capability in discriminating between the participants’ resting state (*nominal*) and active state (*non-nominal*). Such evidences positively answer to the two initial experimental questions, and they pave the path to apply video—based approaches for estimating neurophysiological parameters not only for the telemedicine and remote healthcare, where it would provide a valuable monitoring tool for early adverse clinical events detection [64] especially in pandemic conditions, but also to the industrial automation field, future safety-oriented [62] and operative applications [65]. To this regard, further studies will aim at better investigating the video–based technique sensibility in mental workload and attention discrimination and to determine the optimal application conditions, i.e., the distance between the webcam and the subject’s face, in terms of reliability. Moreover, the combination of both HR and EBR for estimating the above mentioned mental states will be explored, since the merge of these neurophysiological parameters could lead to a more accurate mental workload and attention evaluation [66,67].

## Figures and Tables

**Figure 1 sensors-21-01607-f001:**
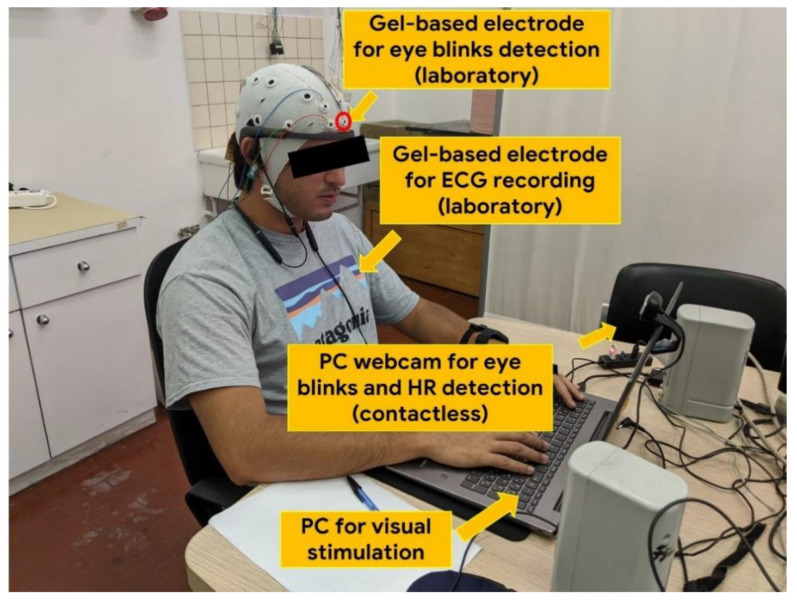
Overview of the experimental settings. Laboratory and video–based equipment was employed to address the objectives of the study. Other acquisition devices were present although they were not used for the purposes of this study.

**Figure 2 sensors-21-01607-f002:**
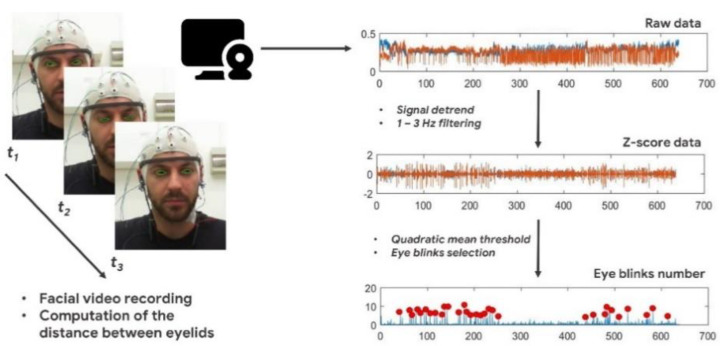
Main steps of the video-signal processing for Eye Blinks Rate (EBR) estimation. The distance between the eyelids is computed frame by frame. Then, filtering and the quadratic mean threshold are applied for obtaining the eye blinks number from the raw data.

**Figure 3 sensors-21-01607-f003:**
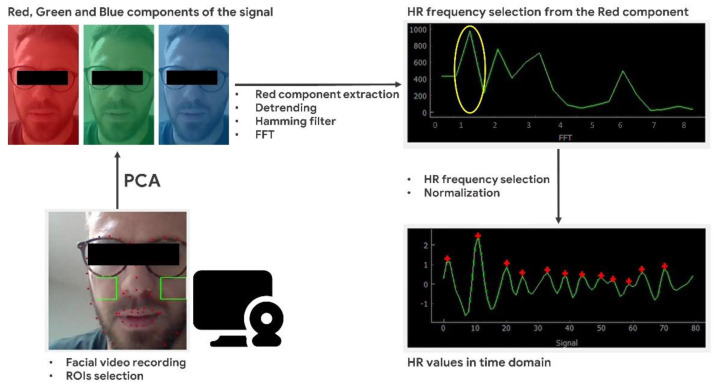
Main steps of the video-signal processing for heart rate (HR) estimation. Starting from the bottom left, the facial video is recorded by mean of a PC webcam and the regions of interest (ROIs) are selected. Then, the R, G and B components are selected by mean of Principal Component Analysis (PCA) algorithm. The Heart Rate (HR) frequency is extracted after detrending, filtering and fast Fourier transformation. Finally, the HR values in time domain are obtained after z-score normalization.

**Figure 4 sensors-21-01607-f004:**
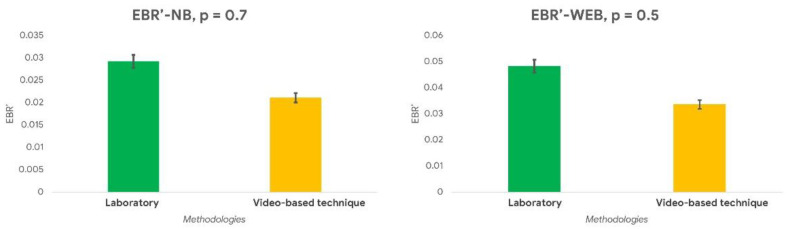
The normalized EBR (EBR’) values evaluated through the video—based technique and the laboratory sensor during the n-Back (NB) (**left** image) and the Webcall (WEB) (**right** image) tasks did not statistically differ (all *p* > 0.05).

**Figure 5 sensors-21-01607-f005:**
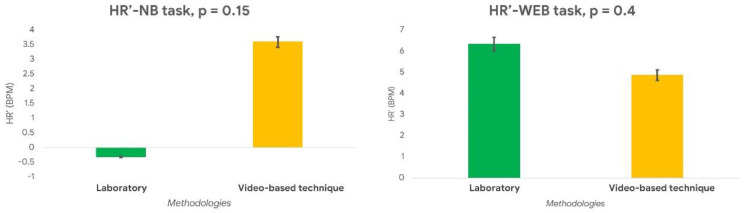
The normalized HR (HR’) values evaluated through the video—based technique and the laboratory sensor during the NB (**left** image) and the WE (**right** image) tasks did not statistically differ (all *p* > 0.05).

**Figure 6 sensors-21-01607-f006:**
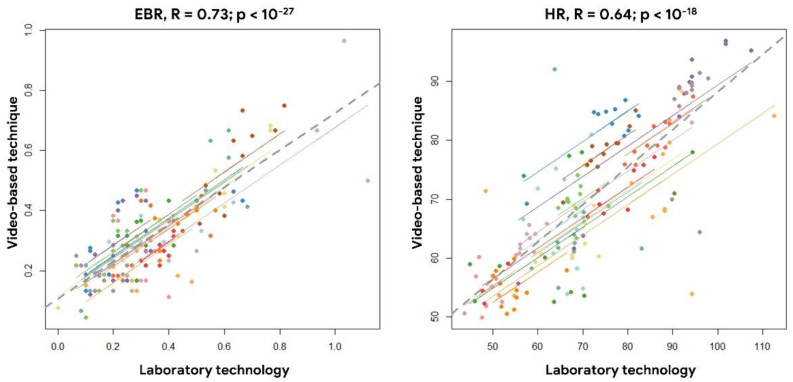
Results of the repeated measure correlation analysis on the EBR (**left** image) and HR (**right** image) estimated by the laboratory and video-based technique every 60 s.

**Figure 7 sensors-21-01607-f007:**
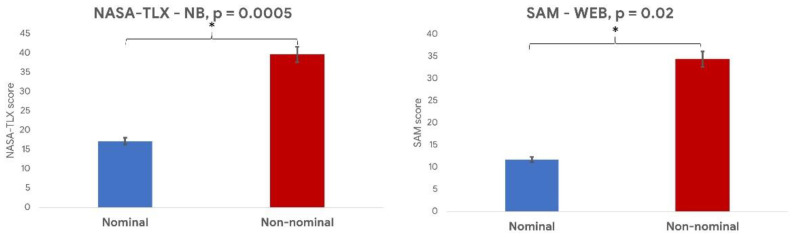
The average NASA-TLX scores during the nominal (blue bar) and non-nominal (red bar) conditions (**left** image), and the average Self Assessment Manikin (SAM) scores in terms of arousal during the nominal and non-nominal conditions (**right** image). * indicates a statistical difference between the represented parameters.

**Figure 8 sensors-21-01607-f008:**
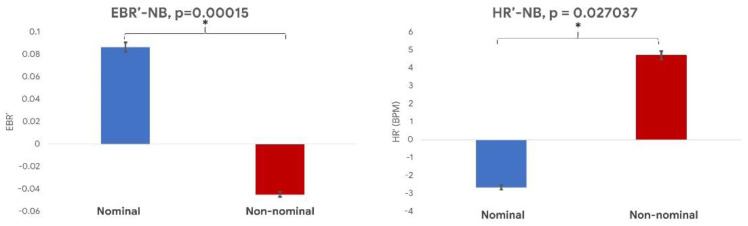
The normalized EBR (**left** image) and HR (**right** image) values during the nominal (blue bar) and the non-nominal (red bar) conditions of the NB task. * indicates a statistical difference between the represented parameters.

**Figure 9 sensors-21-01607-f009:**
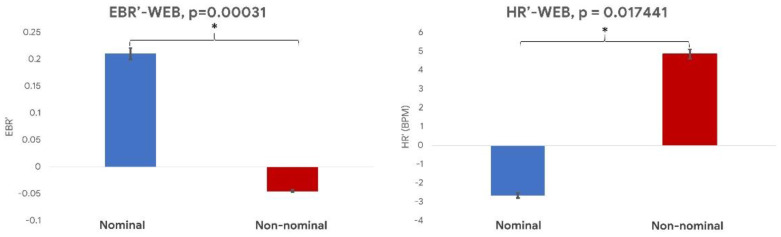
The normalized EBR (**left** image) and HR (**right** image) values during the nominal (blue bar) and the non-nominal (red bar) conditions of the WEB task. * indicates a statistical difference between the represented parameters.

## Data Availability

The aggregated data presented in this study might be available on request from the corresponding author. The data are not publicly available because they were collected within the EU Project “WORKINGAGE: Smart Working environments for all Ages” (GA n. 826232) and they are property of the Consortium.

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
