# Peer review of "A Video-Based Technique for Heart Rate and Eye Blinks Rate Estimation: A Potential Solution for Telemonitoring and Remote Healthcare"

_sensors, 2021, doi:10.3390/s21051607_

Round 1

Reviewer 1 Report

The paper proposes an innovative video-based solution for monitoring potential patients' neurophysiological parameters and assessing their mental state. Using a video-based technique were estimated the Heart Rate (HR) and Eye Blinks Rate (EBR) of 15 participants while performing different tasks.

The results demonstrated the reliability of the innovative video-based technique for computing the EBR and HR neurophysiological parameters.

The scientific level is suitable for a scientific article published in a quality journal. The paper meets all the requirements for publishing.

Reviewer 2 Report

This study examins in healthy volunteers the possibility of remote "diagnosis" without any direct sensor with focus on parallel heart rate and eye blink monitoring (in comparison with "classical" direct devices). Moreover, it aims to test if this method is able to differentiate two different emotional state. The idea is very interesting (although not new, as it is clearly described in introduction), and I see many potential possibility of use. It did not became clear, however, if any outcome of the combined HR and EBR monitoring is superior to any of them alone. Would it be possible to create a combined parameter from both? Both the introduction and discussion is full with possible utility of this method. I suggest to combine them and put it in the discussion, while leaving previous work on the field in the introduction.

The manuscript was well written, was easy to follow. However, from the method section the description of the statistical analysis is missing and the SD or SEM values are also missing from the figures.

I have some further questions/comments.

L 144 Were there any differnce between the difficulity level of the task or positive or negative valance? I am OK with reporting only the average, but I think this information is indeed important even if it explained only in a sentence.

L166 a dot is missing

L174 were placed

L178 delete "one"

L203 "Error! Reference source not found." ???

L213 The electrode is not visible on the Figure.

L220 What does "they" mean here?

L228 "the participant’s facial video," can be deleted

L232 "gathered within" ?

L243-244 explain HR and ROI

L256 The subjective measures have some meaning here only in comparison to objective measures, thus, I would not put these data in a separate chapter.

L287 " p < 10-27" ???

L294 Should be numbered as before. Here this values should have been correlated with the subjective measures.

L335 double denial

L350 Start a new paragraph or even a subchapter with Limitation

L362 Conclusion should be shortened. In its present for a summary, already written in Discussion.

Reviewer 3 Report

The authors investigated the possibility to estimate the Heart Rate (HR) and Eye Blinks Rate (EBR) of participants while performing laboratory tasks by means of facial–video analysis, and the capability of the video-based technique in discriminating between the participant’s resting state (Nominal condition) and active state (Non-nominal condition). Experimental results demonstrate that the HR and EBR estimated through the facial-video technique or the laboratory equipment did not statistically differ and that these neurophysiological parameters allowed the discrimination between the Nominal and Non-nominal states. However, the research method and experimental results are ambiguous and should be clarified explicitly. Furthermore, there are many issues and errors which need to be addressed and are shown as follows.

  1. In section 2.2 “Experimental Protocol”, the authors used the NB task to simulate non-nominal states, demonstrating the normalized HR and EBR of the non-nominal state in Figure 8. Three conditions (0-back, 2-back, and 2-back stressful) of such task were tested. However, considering that the n values obviously affect the workload of the participants, the effect of different n values should be described, and the experimental results of these three conditions have to be discussed separately. Additionally, the authors should also explain the baseline of the NB task, and the reasons for not using 1-back.

  1. The adaBoost classifier played an important role in this work since it was used to automatically identify the participants’ faces. However, the details about the adaBoost classifier are missing in this article, including the training method, the source of the training data, and the purpose of using the classifier.

  1. In section 2.4 “Eye Blinks Signal Recording and Analysis”, the authors used the eye blinks detection method and the equation in line 182 to obtain the ground truth of EBR. To ensure the ground truth is correct and can be used for validation in this experiment, the authors should evaluate that the accuracy of method is comparable to other recognized methods.

  1. In line 171, the authors stated that “The EBR information were obtained by estimating the vertical Electrooculographic (EOG) activity from a traditional Electroencephalography (EEG) channel [R] [41]”. However, there is no explanation for using estimated EOG instead of a directly measured EOG signal.

  1. The “Std-threshold” mentioned in line 183 is not used in the equation. If the “Std-threshold” is 3 in the equation, the authors should state the meaning and usage of the value, and why it is determined to be 3?

  1. In section 2.2 “Experimental Protocol”, three different types of tasks (NB, DG, and WEB) were performed to simulate the non-nominal stat. However, the authors provided no references of DG and WEB to show that they can be used to manipulate workload. Therefore, related references of the two tasks should be provided to ensure they are suitable for this experiment.

  1. Several errors in this article need to be corrected. The equation in line 182 needs to be numbered. The capital of Figure 2 is mistakenly written as "Figure 1". "WEB" is mistakenly written as "WE" in lines 316 and 324. The p value in the left image of Figure 4 does not match the description in line 258. The title “Mental States discrimination” in line 294 needs to be numbered.

Reviewer 4 Report

The authors of this paper said: “The present study explored an innovative approach for the telemedicine and telemonitoring that aims at estimating the Heart Rate (HR) and the Eye Blinks Rate (EBR) through the analysis of the patient’s face video recorded by mean of a webcam”. But, regarding with the state of the art, which are the references [24-28]. It is not clear what the scientific contribution of this article is. 

There are two visible syntax errors:

1.- Line 203, Error! Reference source not found.

2.- Reference [25], line 454, is not complete.

Some comments are listed below:

  1. The system implementation is not mentioned clearly. A broad technical description about the implementation is really necessary to be able to reproduce the experiments.
  2. In methodology section, regarding the computational algorithms to estimate HR and EBR appropriate explanations should be added for realizing better and/or easier understanding of readers. Additional mathematically explanations are required to be added.
  3. The authors should add some statement regarding with scientific contribution to support the viewpoint given in the introduction.
  4. It is not clear how much sensing time and processing time are necessary to estimate a HR and EBR. This is really important because while the system is measuring, the person may change his/her posture some times.
  5. The authors should add the expression of the percentage error in the full range, especially in the conclusion section.

Round 2

Reviewer 4 Report

Thank you for taking my feedback, the authors have worked hard to improve the article.

There are some visible syntax errors:

1.- Line 273, Error! Reference source not found.

2.- Line 277, Error! Reference source not found.

3.- Line 284, Error! Reference source not found.

4.- Line 287, Error! Reference source not found.